# Non-Trivial Dynamics in the FizHugh–Rinzel Model and Non-Homogeneous Oscillatory-Excitable Reaction-Diffusions Systems

**DOI:** 10.3390/biology12070918

**Published:** 2023-06-27

**Authors:** Benjamin Ambrosio, M. A. Aziz-Alaoui, Argha Mondal, Arnab Mondal, Sanjeev K. Sharma, Ranjit Kumar Upadhyay

**Affiliations:** 1UNIHAVRE, LMAH, FR-CNRS-3335, ISCN, Normandie University, 76600 Le Havre, France; 2The Hudson School of Mathematics, New York, NY 10001, USA; 3Department of Mathematics, Sidho-Kanho-Birsha University, Purulia 723104, India; arghamondalb1@gmail.com; 4Department of Mathematical Sciences, University of Essex, Wivenhoe Park, Colchester CO4 3SQ, UK; 5Department of Mathematics and Computing, Indian Institute of Technology (Indian School of Mines), Dhanbad 826004, India; arnab.17dr000410@am.ism.ac.in (A.M.);

**Keywords:** FitzHugh–Nagumo, FitzHugh–Rinzel model, fast-slow dynamics, bifurcation, canard, mixed mode oscillations, bursting oscillations, neuroscience, waves

## Abstract

**Simple Summary:**

This article provides original numerical and mathematical insights about the FHR model and non-homogeneous FitzHugh–Nagumo reaction-diffusion systems.

**Abstract:**

This article focuses on the qualitative analysis of complex dynamics arising in a few mathematical models in neuroscience context. We first discuss the dynamics arising in the three-dimensional FitzHugh–Rinzel (FHR) model and then illustrate those arising in a class of non-homogeneous FitzHugh–Nagumo (Nh-FHN) reaction-diffusion systems. FHR and Nh-FHN models can be used to generate relevant complex dynamics and wave-propagation phenomena in neuroscience context. Such complex dynamics include canards, mixed-mode oscillations (MMOs), Hopf-bifurcations and their spatially extended counterpart. Our article highlights original methods to characterize these complex dynamics and how they emerge in ordinary differential equations and spatially extended models.

## 1. Introduction

The goal of this article is to shed light on mechanisms related to the emergence of complex oscillations arising in a few dynamical systems derivated from the FitzHugh–Nagumo equations (FHN) [1,2]. A specific model of interest here is a non-homogeneous FitzHugh–Nagumo reaction-diffusion (RD)-type system (Nh-FHN) where a parameter is space-dependent. This model derivates from the classical FHN model, which is itself a simplified version of the Hodgkin–Huxley (HH) model [3]. The HH reaction-diffusion model is fundamental in the field of mathematical neuroscience because it stands for the first model generating action potentials through ionic fluxes dynamics and because it allows for their propagation along the axon. Before digging into a more detailed description of the model under consideration here, it is worth providing some elements of context about recent contributions that illustrate how dynamical systems have become a crucial theoretical tool to describe the time evolution of neuronal activity and dynamical brain states. In [4], the authors show how the Drosophila central complex, a region implicated in goal-directed navigation, performs vector arithmetic. The authors describe a neural signal in the fan-shaped body that tracks the allocentric traveling angle of a fly. Previous work had identified neurons in Drosophila that track the heading angle. On this basis, they characterize a neuronal circuit that performs a coordinate transformation and vector addition to compute the allocentric traveling direction. For modeling purposes, the authors use a network of dynamical systems where traveling directions show up as bumps of activity representing calcium concentrations. In [5,6], the authors propose a network model of excitatory and inhibitory neurons to describe the activity of the visual cortex. Their work highlights emergent properties such as Gamma rhythms. It also illustrates how the network, doing its own mathematical computations, mimics the visual cortex response to orientated visual stimuli. Again, they claim that the network dynamical system, and the brain by extrapolation, computes mathematical convolutions to deliver cortical responses. Finally, in [7], Volpert et. al. investigate the spatiotemporal dynamics of neural oscillations observed in real EEG data acquired during a picture-naming task. Their theoretical models rely on one hand on analytical solutions of non-autonomous Poisson equations and on the other hand on optical flow patterns such as sources, sinks, spirals and saddles. Their work contribute to a better understanding of the neural dynamics at the macroscopic scale with the objective to characterize neural activity corresponding to a specific behavior. Overall, all these works consecrate the use of Dynamical Systems to characterize neural activity in a specific context. They appeal for a better knowledge of the mechanisms inducing oscillations, neural rhythms and spatiotemporal organization. This list of relevant works could be extended widely. Here, our focus is on how complex oscillations emerge and propagate in FitzHugh–Nagumo inspired systems. This question arises naturally in the context of propagation of an action potential along an axon or in excitable media. In recent decades, a significant progress relying on the properties of FHN type models has been made in the theoretical description of cardiac tachycardia and fibrillation, see [8,9] and references cited therein. For axonal propagation, the article [10] provides a typical example of a phenomenological study relying on excitable models. In this article, the authors indeed rely on a non-homogeneous FHN RD-type system to give numerical insights about how to distinguish between geometrical enlargements that lead to minor changes in propagation from those that result in critical phenomena such as blockages of the original traveling spike. They focus specifically on the inhomogeneity in the diffusion’s coefficient to take into account the discrepancy of the axonal diameter. Our approach is by several aspects related to the contribution of [10], but in a more theoretical framework. We seek to highlight the detailed dynamical mechanisms that can shape complex oscillations and affect their propagation across spatial domains. Coming back to the Nh-FHN model, the space domain, typically stands for an excitable tissue such as a neural tissue or a neuron’s axon. One interesting property of the FHN ODE is that upon variation of a parameter, a globally attracting limit-cycle emerges from an Hopf-bifurcation. Depending on the value of this parameter, FHN can be either excitable or oscillatory. For the specific case of Nh-FHN, at a designated restricted location of the spacial domain, the cells are assumed to be in an oscillatory state, and elsewhere in the domain, the cells are excitable, see Figure 1. The excitability of the cells depend on the value of the aforementioned parameter, which now depends on the space variable, and which in application is related to the external current. In this context, for Nh-FHN, we observe the following phenomenon; upon the variation of this parameter value related to excitability, waves of depolarization may propagate from the center of the domain toward the boundaries when the cells are enough excitable, or the solution can evolve to a stationary state if the cells are not enough excitable. For a parameter value range in between, a bifurcation occurs, typically a Hopf-bifurcation. Within this range of parameters, for some cells, mixed mode oscillations or other complex dynamics can be observed in time. This scenario was first described in [11] and further analyzed and discussed in a few papers such as [12,13]. Other scenarios can be drawn with Nh-FHN, for example one could include damaged tissue parts in the domain, see Figure 1. Numerical simulations have been developed for this case, but the details of it are beyond the scope of the present work. The wave propagation phenomenon in excitable media and in FHN RD in particular has been intensively investigated for a few decades, see for example [14,15,16,17,18,19,20,21] and references therein. Since the seminal study of the Fisher KPP equations, a general theory of traveling waves has also been developed, see for example [22,23,24]. The numerical simulations to be presented in this article suggest, however, that the vast literature does not account for the details of the dynamics illustrated here. Our focus here is to present few oscillatory complex phenomena that appear to be tractable by analogy with lower dimensional systems. Our approach is to emphasize the analogy between some dynamics observed in the spatial Nh-FHN system for some cells and the dynamics of the FHR ODE system. This idea gives the structure of the present article. After the introductory part, we provide an original theoretical and numerical analysis of the FHR system in Section 2. We then illustrate numerical simulations and theoretical results for Nh-FHN in Section 3. Before digging into a more detailed analysis, we first introduce the equations to be considered, recalling also some necessary background.

## 2. Methods

### 2.1. The FHR System

The FHR ODE model was introduced as a simple model to generate bursting oscillations, (i.e., alternated phases of oscillations and quiescent states) in [25,26], and previously studied by J. Rinzel and R. FitzHugh in an unpublished paper in 1979. In those fundamental papers, the FHR model reads as
(1)$$\left\{\begin{array}{l} \epsilon \frac{dv}{dt}=f\left(v\right)-w+y+I,\\ \;\frac{dw}{dt}=\phi \left(a+v-bw\right)\\ \;\frac{dy}{dt}=\epsilon \left(c-v-dy\right) \end{array}\right.$$
with
$$f\left(v\right)=-\left(1/3\right)v^{3}+v$$

For the following values of parameters, I=0.3125, a=0.7, b=0.8, c=−0.775, d=1.0,ϕ=0.08,ϵ=0.0001 the model exhibits nice bursting behavior. The mechanism is strikingly simple and intuitive. The model consists of a FitzHugh–Nagumo (FHN) system, represented by the two first equations and a super slow variable, whose dynamics are given by the third equation. A relevant approach here is to formally consider y+I as a parameter for the two first equations. Then, the dynamics of the two first equations are known from the FHN analysis. But the variable *y* moves slowly with the variation of the first variable, allowing the emergence of the bursting. The description made in the original papers [25,26] reflects a precise numerical qualitative analysis of the FHR dynamics supported by numerical illustrations. The author suggested that the model would be suitable for a deeper rigorous mathematical analysis, and invited the community to study the model in this direction. This call would receive important feedback, and the model was indeed studied later in numerous papers, see for example [27,28,29,30] to cite only but a few. More generally, the idea of adding one or more variables with different scales to capitalize on the known dynamics of FHN was successful in many aspects. For example, in [31], see also [32], the dynamics of FHN at different scales were used to model the alternating pulse and surge pattern of gonadotropin-releasing hormone secretions. In [33], a three-dimensional FHN served as an explanatory example to illustrate the inflection sets in the context of the excitability threshold, a method previously developed for planar systems, see [34] and references therein. In particular, such 3D systems are reference models for the emergence of complex oscillations such as MMOs and bursting, which have been widely observed in biological contexts [31,35,36,37,38]. The analysis of systems with multiple scales is a very active topic of research with numerous applications in biology, physics and chemistry, see [39,40,41]. Here, we will consider a slightly different version of the original FHR, for which we will provide an original numerical and theoretical analysis. The system under consideration writes: (2)$$\left\{\begin{array}{ccc} \epsilon \frac{du}{dt} & = & f(u)-v+w+I,\\ \frac{dv}{dt} & = & u-bv-c\\ \frac{dw}{dt} & = & \epsilon (-u-w) \end{array}\right.$$
with *f* a cubic function, b>0, ϵ a small parameter and *I* a parameter to vary. For the ODE section, we will provide an analysis with *f* and *b* set as
$$f\left(u\right)=-\left(1/3\right)u^{3}+u,b=0.8,c=0.$$
Next, we move forward with a short presentation of the Nh-FHN model.

### 2.2. A Non-Homogeneous FHN Model (Nh-FHN)

The non-homogeneous FHN reaction-diffusion model (Nh-FHN) to be considered here writes
(3)$$\left\{\begin{array}{l} \epsilon \frac{du}{dt}=f\left(u\right)-v+w+I\left(x\right)+d_{u}\Delta u,\\ \;\frac{dv}{dt}=u-bv-c\left(x\right)+d_{v}\Delta v \end{array}\right.$$
on a real bounded interval space domain (α,β) or in a two dimensional square with Neumann Boundary Conditions (NBC). The notation I(x) and c(x) are used on purpose to emphasize their dependence in the space variable. The notations Δu stands for $\sum_{i=1}^{n}\frac{\partial^{2}u}{\partial x_{i}^{2}}$. Around the center of the interval (α,β), I(x) and c(x) are set to a value for which the diffusion-less ODE underlying system would generate relaxation oscillations. Out of this region, *I* and *c* are set to a value for which the diffusion-less ODE would be in a stationary stable but excitable state. We allow those functions to be varied as a function of a parameter; this leads to a bifurcation path from a stationary state to propagation of oscillations for Nh-FHN. Previously, various studies have been conducted by some of the authors of the present paper, see [11,12,13]. In Section 4, we will present numerical simulations of this PDE to be compared to the dynamics of Equation (2). We will consider the following set of parameters in Section 4.1
$$\begin{array}{c} n=2,d_{u}=d_{v}=1,b=0,f\left(u\right)=-u^{3}+3u\\ I(x)=0\\ c(x)=0\;\;\mathrm{in}\;a\;\mathrm{small}\;\mathrm{ball}\;\mathrm{at}\;\mathrm{the}\;\mathrm{center}\\ c\left(x\right)=c_{0}=-1.21\;\;\mathrm{otherwise} \end{array}$$
and also
$$\begin{array}{c} n=1,\Omega =\left(\alpha ,\beta \right),d_{u}=d=1,d_{v}=0,b=0,f\left(u\right)=-u^{3}+3u\\ I(x)=0\\ c\left(x\right)=p{\left(x/\beta \right)}^{4}-2p{\left(x/\beta \right)}^{2}\\ \beta >0,\;\alpha =-\beta \end{array}$$
in Section 4.2. All the partial differential equations will be solved with a RK4 discretization in time and a finite difference scheme in space.

## 3. Analysis of the FHR System

### 3.1. A Short Background on FHN

The mechanisms at play in the FHR and Nh-FHN models rely primarily on the dynamics of the FHN model. Consequently, it is worth to first briefly describe the dynamics of the following classical FHN system:(4)$$\left\{\begin{array}{ccc} \epsilon \frac{du}{dt} & = & f(u)-v+I\\ \frac{dv}{dt} & = & u-bv, \end{array}\right.$$
where $f\left(u\right)=-\frac{u^{3}}{3}+u$, ϵ is a small parameter, 0<b<1 an I>0.

**Proposition 1.** 
*Equation (4) admits a unique stationary solution $\left(u^{*},v^{*}\right)$ given by*

$$v^{*}=u^{*}/b$$

*and $u^{*}$ is the unique solution of*

f(u)−u/b+I=0.

*It follows that $u^{*}$ is an increasing function of I; and the map $I\to u^{*}$ is a bijection from (0,+∞) to (0,+∞). Furthermore, as I increases from 0 to +∞, $\left(u^{*},v^{*}\right)$ is successively a repulsive node, a repulsive focus, an attractive focus and an attractive node. At $f^{\prime }\left(u^{*}\right)=b\epsilon \;\;\;\left(i.e.,\;\;u^{*}=\sqrt{1-b\epsilon }\right)$ a Hopf bifurcation occurs.*


Furthermore, when $u^{*}$ is close to zero, the system is known to exhibit relaxation oscillations. With this result in mind, one principle at play to obtain MMOs becomes clear: if we add a third variable that moves slowly, and in such a way that the dynamics for the system of the two first equations follow the loop: attractive focus, repulsive focus, relaxation oscillation and then return mechanism, MMOs should appear. The next subsection illustrates in more detail this idea to obtain MMOs as well as other phenomena related to the so-called canard solutions.

#### 3.1.1. A System with MMOs

We consider here the Equation (2) introduced above
$$\left\{\begin{array}{ccc} \epsilon \frac{du}{dt} & = & f(u)-v+w+I,\\ \frac{dv}{dt} & = & u-bv\\ \frac{dw}{dt} & = & \epsilon (-u-w) \end{array}\right.$$
with
$$f\left(u\right)=-\left(1/3\right)u^{3}+u,\epsilon =0.1,\;I\in \left(1.3,1.5\right),b=0.8.$$
We focus on this system because it exhibits MMOs. We describe here two distinct mechanisms leading to alternance of small and large oscillations. The first mechanism corresponds to small oscillations related to the focus nature of the fixed point in the two-dimensional FHN Equation (4) as described in Section 1. The second mechanism is different; in this case, the trajectories clearly exhibits canard-type solutions, and the trajectories jump from the repulsive manifold toward alternatively one side or the other side of the stable manifold. Of note, this distinctive evolution toward the left or right side of the stable manifold occurs within a tiny region of the repulsive manifold. We end the section with a remark on Shilnikov Chaos.

#### 3.1.2. MMOs and Focus

The first mechanism to obtain MMOs is as follows. Considering *w* as a parameter, the two first equations represent a classical FHN system, which goes through a Hopf bifurcation as *w* is varied. If *w* varies very slowly, in an interval corresponding to a focus for the FHN system, then we will observe focus-like dynamics in a neighborhood of the fixed point of the FHN subsystem. The focus is first attractive, then repulsive, until the trajectory falls into a relaxation oscillation type. This corresponds to small oscillations followed by a large oscillation. The dynamics are such that during the relaxation oscillation *w* returns close to its initial value. This is a return mechanism, from which a new cycle follows. This behavior is illustrated in Figure 2.

**Remark 1.** 
*Another approach to describe the above dynamics relies on canards. The small oscillations are also to be seen as trajectories successively following the attractive and repulsive parts of the manifold till they exit the vicinity of the fold through a relaxation oscillation. We wanted here to emphasize an approach by a dynamical Hopf bifurcation, which appears to be relevant.*


#### 3.1.3. MMOs and Canards

The second mechanism we want to discuss gives rise to a quite different type of dynamics. In this case, there are no multiple small focus-like oscillations. Instead, after a relaxation oscillation, the trajectory may enter a canard-type trajectory after crossing the apex of the right part of the critical manifold; it follows the unstable manifold and leaves it either to the left side—where it reaches the left part of the attractive critical manifold—or to the right side—and reaches the right part of the attractive manifold. The latter case corresponds to a small (or middle) oscillation while the first case gives a large relaxation oscillation. This is illustrated in Figure 3. It is worth to describe in more detail the trajectory represented in this figure. Two large oscillations are followed by a single small oscillation where the trajectory follows the repulsive manifold before being attracted by the right side of the attractive manifold. This is repeated four times, the fifth time, the second large oscillation is replaced by a small oscillation (but larger than the other small oscillations). This corresponds to a brutal change of direction in the trajectory occurring in a tiny zone of the phase space. After that, the cycle is repeated. Introducing the letter M for the medium oscillation, L for large and S for small, we could denote this occurrence: LLS-LLS-LLS-LLS-LM. Note that during successive cycles LLS, the canards occurring during the second L, are significantly different in the phase space, and move forward in a specific direction. At the forth time the canard exits the unstable manifold to the right. Of note, this discrimination between left and right exits has been used to generate rich dynamics in various contexts, see the recent papers [27,42].

**Remark 2.** 
*In the next section, computations will show that the regimes of MMOs exhibited below correspond to a range of parameter value I for which the fixed point has two complex eigenvalues with a positive real part and one negative real value. This is a signature of Shilnikov Chaos, which relates to dynamics alternating phases on the attractive manifold and phases on the repulsive manifold supported by the complex eigenvalues [43,44] and references therein. Figure 4 gives an interesting glimpse of it. For the initial condition considered here, at the beginning, the solution follows the stable manifold corresponding to the negative eigenvalue. But afterwards, the trajectory exits the neighborhood of the fixed point with a focus-like dynamics; this corresponds to the two complex eigenvalues with positive real parts. After that, the trajectory never comes back in a neighborhood of the fixed point— the trajectory following the stable manifold corresponds to a transient behavior—and as such the asymptotic observed dynamics do not relate to Shilnikov chaos despite the eigenvalues.*


### 3.2. Basic Stability Analysis

The following proposition results from simple computations.

**Proposition 2.** 
*For any b>0, Equation (2) admits a unique fixed point $\left(u^{*},v^{*},w^{*}\right)$ given by*

$$v^{*}=u^{*}/b,\;\;w^{*}=-u^{*}$$

*where $u^{*}$ is the unique solution of*

$$f\left(u\right)-\left(1+\frac{1}{b}\right)u+I=0.$$



A local stability analysis provides the following proposition.

**Proposition 3.** 
*There exists $I^{*}\in \left(1,2\right)$ such that at $I=I^{*}$, an a Hopf bifurcation occurs.*


**Proof.** The Jacobian matrix J(u) writes
(5)$$\left(\begin{array}{ccc} \frac{f^{\prime }\left(u\right)}{\epsilon } & -\frac{1}{\epsilon } & \frac{1}{\epsilon }\\ 1 & -b & 0\\ -\epsilon & 0 & -\epsilon \end{array}\right)$$
which gives
$$Det\left(J\left(u\right)-\lambda I\right)=-\lambda^{3}+\lambda^{2}\left(-\epsilon -b+\frac{f^{\prime }\left(u\right)}{\epsilon }\right)+\lambda \left(f^{\prime }\left(u\right)+b\frac{f^{\prime }\left(u\right)}{\epsilon }-b\epsilon -\frac{1}{\epsilon }-1\right)+bf^{\prime }\left(u\right)-1-b$$
$$=-\lambda^{3}+\lambda^{2}\left(-\epsilon -b+\frac{f^{\prime }\left(u\right)}{\epsilon }\right)+\lambda \left(f^{\prime }\left(u\right)+b\frac{f^{\prime }\left(u\right)}{\epsilon }-b\epsilon -\frac{1}{\epsilon }-1\right)-bu^{2}-1$$
Thanks to simple algebraic computations (Routh–Hurwitz criterion and Cardan formula), one can prove that in the interval (1,2) an eigenvalue is real negative and the two other are complex conjugate; moreover, for $I<I^{*}$, the stationary point is unstable, and for $I>I^{*}$, it is stable. □

See Figure 5 for numerical illustration.

### 3.3. Absorbing Set, Existence of Periodic Solution, Numerical Approximation

We assume ϵ>0 is small enough.

**Proposition 4.** 
*System (2) admits an absorbing bounded set.*


**Proof.** We define
$$\psi \left(t\right)=\epsilon u^{2}+v^{2}+w^{2}$$
We have
$$\psi^{\prime }=-\frac{u^{4}}{3}+u^{2}+\left(1-\epsilon \right)uw+Iu-bv^{2}-\epsilon w^{2}$$
By using young inequalities, we can prove that there exist two positive constants $K_{1}$ and $K_{2}$ such that
$$\psi^{\prime }\leq -K_{1}g+K_{2}.$$
Multiplying both sides by $e^{K_{1}t}$ and integrating leads to
$$\psi \left(t\right)\leq e^{-K_{1}t}g_{0}+\frac{K_{2}}{K_{1}}\left(1-e^{K_{1}t}\right).$$
This completes the proof. □

**Theorem 1.** 
*There exists $I_{0}$, such that for $I\in [0,I_{0})$, the system admits a non-constant periodic solution.*


**Proof.** We assume I=0. The result extends to $\left[0,I_{0}\right)$ by continuity. We consider the Poincaré map $F=(F_{1},F_{2},F_{3})$ from the manifold M={(u,v,w);u=0,v<0,w∈R} to itself defined thanks to the flow of the ODE. For any *w* such that |w| is not too large, and for ϵ is small enough, the map is well-defined thanks to the slow-fast theory since the trajectories are close to relaxation oscillations and $w^{\prime }$ is of O(ϵ). We know also that the trajectories will return at M with a v–coordinate at f(−1)+O(ϵ) (with *f* defined in Equation (2), and since $f^{\prime }\left(-1\right)=0$, see classical works on slow-fast systems such as for example [45]). From the Brouwer theorem, we deduce that for each fixed *w*, there exists *v* such that
$$F_{2}\left(0,v,w\right)=v$$
i.e., the *v*-coordinate is a fixed point. By this way, we define a continuous function w→φ(w)=v such that
$$F_{2}\left(0,\varphi \left(w\right),w\right)=\varphi \left(w\right).$$
Next, we look for $w^{*}$ such that,
$$F_{3}\left(0,\varphi \left(w^{*}\right),w^{*}\right)=w^{*}.$$Let us start with an initial condition $w_{0}<0$ close to 0. We have
$$F_{3}\left(0,\varphi \left(w_{0}\right),w_{0}\right)=w_{0}e^{-\epsilon T}-\epsilon \int_{0}^{T}e^{-\epsilon (T-s)}u\left(s\right)ds$$
where *T* is the time at which the flow ensued from $\left(0,\varphi \left(w_{0}\right),w_{0}\right)$ returns to M. Replacing the exponential by its order 1 Taylor expansion, we found that adopting the notation $F_{3}\left(w_{0}\right)$ for the sake of simplicity
$$F_{3}\left(w_{0}\right)=w_{0}-\epsilon w_{0}T-\epsilon \int_{0}^{T}u\left(s\right)ds+o\left(\epsilon \right)$$
which gives $F_{3}\left(w_{0}\right)>w_{0}$, for $w_{0}<0$ in a small neighborhood of 0 and ϵ small enough. Analog arguments allow us to prove that $F_{3}\left(w_{0}\right)<w_{0}$ for $w_{0}>0$ and ϵ are small enough. We omit here the details of the computations that can be made explicit by changing the variable of integration from *s* to *u* and rely on the fact that the time spent on the right or left part of the stable manifold depends on the sign of *w*; this corresponds geometrically to the relative position of the nullclines. It follows that there exists $w^{*}$ such that $F_{3}\left(w^{*}\right)=w^{*}$ and therefore $\left(0,\varphi \left(w^{*}\right),w^{*}\right)$ is a fixed point of the Poincaré map. □

### 3.4. A Numerical Approximation for Small Oscillations

The aim of this section is to illustrate how the small oscillations observed during typical MMOs (as in Figure 2) can be locally captured by the dynamics of a “moving” focus. Observation of the small oscillations of Figure 2 show that oscillations in the (u,v) plane are first decreasing in amplitude, then increasing till the trajectory leaves the vicinity of the fold line. Since *w* moves very slowly, this corresponds to the fact that for fixed *w* in the corresponding range, the stationary point of the two first equations is a focus, first stable and then unstable, see Proposition 1. In this section, we will approximate the dynamics of the full system by the dynamics of a simpler system for which computations can be made quite explicitly. We will first operate a change variables to translate our attention on the dynamics around the focus. Then, we will look at the linearization and compare the dynamics of the simple system with the original one. The next proposition gives the dynamics in a system of coordinates around $\left(u^{*}\left(w\right),v^{*}\left(w\right)\right)$ where $\left(u^{*}\left(w\right),v^{*}\left(w\right)\right)$ is the “stationary” solution of the subsystem in (u,v) considering *w* constant. Since *w* is super slow, this system of coordinates is relevant.

**Proposition 5.** 
*Let $\left(u^{*}\left(w\right),v^{*}\left(w\right)\right)$ be the unique solution of*

(6)
$$\left\{\begin{array}{ccc} 0 & = & f(u)-v+w+I,\\ 0 & = & u-bv \end{array}\right.$$

*Then, after a change of variables around $\left(u^{*}\left(w\right),v^{*}\left(w\right)\right)$, and with appropriate notations, the trajectory of the solutions of (2) is given by:*

(7)
$$\left\{\begin{array}{ccc} \epsilon \frac{du}{dt} & = & g\left(u^{*},u\right)-v+\epsilon^{2}\frac{1}{f^{\prime }\left(u^{*}\right)-\frac{1}{b}}\left(u^{*}+u+w\right),\\ \frac{dv}{dt} & = & u-bv+\epsilon \frac{1}{f^{\prime }\left(u^{*}\right)-\frac{1}{b}}\left(u^{*}+u+w\right)\\ \frac{dw}{dt} & = & -\epsilon \left(w+u^{*}+u\right) \end{array}\right.$$

*with*

(8)
$$g\left(u^{*},u\right)=\left(1-u^{*2}\right)u-u^{*}u^{2}-\frac{u^{3}}{3}$$



Next, our goal is to replace Equation (7) with a simpler system that mimics the small oscillations. Now, since *w* is very slow, when (u,v) is small in Equation (7), it is geometrically relevant to approximate solutions of Equation (7), during the specific period of small oscillations, by the system: (9)$$\left\{\begin{array}{ccc} \epsilon \frac{du}{dt} & = & (1-u^{*2})u-v,\\ \frac{dv}{dt} & = & u-bv\\ \frac{dw}{dt} & = & -\epsilon \left(w+u^{*}+u\right) \end{array}\right.$$
What is interesting with system (9) is that it gives you a practical way to compute the number of oscillations involved. Let
$$A=\left(\begin{array}{cc} \frac{1}{\epsilon }\left(1-u^{*2}\right) & -\frac{1}{\epsilon }\\ 1 & -b \end{array}\right)$$
And let
$$\beta =0.5\sqrt{4\det A-{\left(TrA\right)}^{2}}$$
The following proposition holds.

**Proposition 6.** 
*The number of small oscillations $n_{o}$ occurring during a time interval (0,T) in Equation (9) is given by*

$$n_{o}=\frac{1}{2\pi }\int_{0}^{T}\beta \left(t\right)dt$$



**Proof.** The idea is to work with an approximate solution of Equation (9). We divide the time interval (0,T) into the subdivision
$${\left(kdt\right)}_{k\in \{0,1,2,...,N\}},\;\;\mathrm{with}\;N=\frac{T}{dt}$$Then, at each time step, for fixed *w*, it is possible to compute the solution of the linear subsystem made of the first two equations of (9) on the interval (kdt,(k+1)dt). Eigenvalues are given by
$$0.5(TrA_{-}^{+}\sqrt{{\left(TrA\right)}^{2}-4\det A})$$
and are complex conjugate when
$$1-u^{*2}≃b\epsilon$$
Classically, denoting α+iβ one of this eigenvalues, we have after a new change of variables,
$$\left(\begin{array}{c} \tilde{u}\\ \tilde{v} \end{array}\right)=\left(\begin{array}{cc} \alpha +b & \beta \\ 1 & 0 \end{array}\right)$$
The solution of this last system writes after dropping the tilde
u(t)+iv(t)=exp(αt)exp(iβ)
Now, the number of small oscillations is given by
$$\sum_{k}\frac{dt}{2\pi }\beta_{k}$$
Now, since the approximation converges towards the solution, the results follow from
$$n_{0}=\lim_{N\to +\infty }\sum_{k=0}^{N}\frac{dt}{2\pi }\beta_{k}=\frac{1}{2\pi }\int_{0}^{T}\beta \left(t\right)dt$$ □

**Remark 3.** 
*The interest of this approximation is that it provides a geometrical approach to interpret the dynamics of small oscillations occurring in MMOs: the amplitude of the solutions first decrease and then increase. For system (7), this corresponds to $\alpha (u^{*}\left(t\right))<0$, in which case the amplitude decreases and then to $\alpha (u^{*}\left(t\right))>0$ in which case the oscillations increase.*


**Remark 4.** 
*Additionally, note that system (9) provides a simple way to generate small oscillations and control the number $n_{0}$; this approach captures the essential phenomenon at play in the generation of small oscillations in system (2). Figure 6 illustrates both oscillations for system (9) and (2). System (9) could be extended to generate this small oscillation recurrently with a reset as it is carried out in the classical Leaky Integrate and Fire models often used in applications [5,6,46]. The difference being that in LIF models, the equation is linear and one-dimensional with respect to a variable V (the voltage) with a reset that occurs when a V reaches a threshold value—while other inputs are typically included to drive this variable V. Here, the idea would be to reset the value w when it reaches the desired threshold value. In between the resets, the dynamics are oscillatory in the variables (u,v).*


### 3.5. Slow-Fast Analysis

In this section, we shall give some insights about the dynamics of Equation (2) thanks to a slow–fast approach. Setting ϵ=0 in Equation (2) after different time scalings provides the main dynamical picture. First, setting ϵ=0 gives
(10)$$\left\{\begin{array}{ccc} 0 & = & f(u)-v+w+I,\\ v_{t} & = & u-bv\\ w_{t} & = & 0 \end{array}\right.$$
Then, after the time rescaling $t=\epsilon t^{\prime }$ in Equation (2), dropping the ${}^{\prime }$ and setting ϵ=0 gives
(11)$$\left\{\begin{array}{ccc} u_{t} & = & f(u)-v+w+I,\\ v_{t} & = & 0\\ w_{t} & = & 0 \end{array}\right.$$
Finally, rescaling with $t=\frac{1}{\epsilon }t^{\prime }$, dropping the ${}^{\prime }$ and setting ϵ=0 gives
(12)$$\left\{\begin{array}{ccc} 0 & = & f(u)-v+w+I,\\ 0 & = & u-bv\\ w_{t} & = & -(w+u) \end{array}\right.$$
Equations (10) and (11) can be seen, for a fixed constant *w*, respectively, as the reduced and the layer systems of the classical 2D FHN system. We first consider the fast dynamics. Outside of the critical manifold, the trajectories are given by the layer system (11), which is a one dimensional ODE in *u*. For any initial condition (unless we start at the repulsive point or in the stable manifold of a saddle), the trajectory will reach one of the attractive parts of the critical manifold v=f(u)+w+I, where $f^{\prime }\left(u\right)<0$.

Then, we look at the slow dynamics. Equation (10) is an ODE on the critical manifold. If the stationary point of Equation (10) is on the attractive part of the critical manifold, the solutions are well defined, and they will evolve to it. To further capture the evolution of the system, we need to consider the very slow motion given by Equation (12). If this stationary point is on the repulsive part with $f^{\prime }\left(u\right)>0$, then when it reaches the fold line f(u)=0, system (10) is not defined, and the derivative explodes in finite time; this corresponds to a jump from one side of the attractive part of the critical manifold to the other one. If the stationary point is on the fold line, a more complex behavior (MMOs) can be expected, see [47,48] and references therein. An interesting and less classical insight from slow-fast analysis relates to the emergence of the sequence LLSLLSLLSLLSLM described in Section 3.1.3. In this case, the transition between canards exiting to the left (large oscillation) and canards exiting to the right (middle oscillation) plays a crucial role. Following the ideas in [49,50], we highlight hereafter some relevant computations. Consider Equation (2)
$$\left\{\begin{array}{ccc} \epsilon u_{t} & = & f(u)-v+w+I\\ v_{t} & = & u-bv\\ w_{t} & = & -\epsilon (u+w) \end{array}\right.$$

We use the change of variables:$$u=1+\bar{u},v=\tilde{v}+\bar{v},w=\tilde{w}+\bar{w}$$
with
$$f\left(1\right)-\tilde{v}+\tilde{w}+I=0$$
$$1-b\tilde{v}=0$$
which leads to the equation (dropping the bars)
$$\left\{\begin{array}{ccc} \epsilon u_{t} & = & -3u^{2}-u^{3}-v+w,\\ v_{t} & = & u-bv\\ w_{t} & = & -\epsilon (\mu +u+w) \end{array}\right.$$
with
$$\mu =1+\tilde{w}=-1+\frac{1}{b}-I$$
Next, we proceed to the change of variables
$$u=\sqrt{\epsilon }\bar{u},v=\epsilon \bar{v},w=\epsilon \bar{w}$$
Further applying the change of time
$$t=\frac{\bar{t}}{\sqrt{\epsilon }}$$
and dropping the bars, we obtain
$$\left\{\begin{array}{ccc} u_{t} & = & -3u^{2}-\sqrt{\epsilon }u^{3}-v+w,\\ v_{t} & = & u-b\sqrt{\epsilon }v\\ w_{t} & = & -\sqrt{\epsilon }\left(\mu +\sqrt{\epsilon }u+\epsilon w\right) \end{array}\right.$$
Setting ϵ=0 gives
(13)$$\left\{\begin{array}{ccc} u_{t} & = & -3u^{2}-v+w,\\ \frac{dv}{dt} & = & u\\ \frac{dw}{dt} & = & 0 \end{array}\right.$$

**Proposition 7.** 
*The point (u,v)=(0,w) is a stationary point of center type for the projection of Equation (13) into the (u,v)-plane. Every trajectory passing through (0,c) with $w<c<w+\frac{1}{6}$ is a periodic solution. Every solution passing through (0,c) with $c\geq w+\frac{1}{6}$ satisfies u(t)→−∞ and v(t)→−∞ as $t\to t^{*}$, where $t^{*}$ is the maximal time for which the solution is defined. Furthermore, as c goes to $w+\frac{1}{6}$, the diameter of the periodic solution goes to +∞.*


**Proof.** Without loss of generality, we work with w=0, since the general case follows from the change of variable
$$-\tilde{v}+w=-v$$
Next, we note that the function
$$G\left(u,v\right)=\left(-3u^{2}-v+1/6\right)\exp\left(6v\right).$$
is a first integral of Equation (13) (with w=0). It follows that the solutions of (13) are contained in the level sets of the function *G*. The proposition then follows from quite long yet elementary computations to explicit the details of the level sets of *G*. We avoid the details here. An illustrative picture is provided in Figure 7. □

## 4. Dynamics in the Nh-FHN Model

In this section, we shall consider the following Nh-FHN system:(14)$$\left\{\begin{array}{ccc} \epsilon u_{t} & = & f\left(u\right)-v+I\left(x\right)+d_{u}\Delta u,\\ v_{t} & = & u-bv-c\left(x\right)+d_{v}\Delta v \end{array}\right.$$
on a regular bounded domain $\Omega \subset R^{n},n\leq 3$ with Neumann Boundary Conditions (NBC).

### 4.1. Existence and Uniqueness of the Stationary Solution

As far as we know, the existence result for Equation (14) in the case $d_{u}>0,\;d_{v}>0$ has not been proved. We consider here the case $d_{u}=d>0,\;d_{v}=0$. In this case, the equation writes
(15)$$\left\{\begin{array}{ccc} 0 & = & f(u)-v+I(x)+d\Delta u,\\ 0 & = & u-bv-c(x) \end{array}\right.$$

When b=0 (and $d_{v}=0)$, the existence of a stationary solution is straightforward since Equation (15) becomes u=c,v=f(c)+I+dΔu. We consider hereafter the case b>0.

**Theorem 2.** 
*Equation (15) admits a unique solution.*


**Proof.** Equation (15) is equivalent to
(16)$$\left\{\begin{array}{ccc} \;0 & = & f(u)-(u/b)+(1/b)c(x)+I(x)+d\Delta u,\\ \;v & = & \left(1/b)(u-c(x)\right) \end{array}\right.$$
Then, we apply the result in [51]. □

### 4.2. Numerical Experiments

The numerical simulations presented in this paragraph have been performed using our own C++ program.

#### 4.2.1. Filtering of Frequencies and Local Mixed Mode Oscillations (MMOs)

In this paragraph, we shall discuss some qualitative dynamics arising near a Hopf-bifurcation. We illustrate the numerical simulation of system (14) with the following set of parameters:(17)$$\begin{array}{c} n=2,d_{u}=d_{v}=1,b=0,f\left(u\right)=-u^{3}+3u\\ I(x)=0\\ c(x)=0\;\;\mathrm{in}\;a\;\mathrm{small}\;\mathrm{ball}\;\mathrm{at}\;\mathrm{the}\;\mathrm{center}\\ c\left(x\right)=c_{0}=-1.21\;\;\mathrm{otherwise} \end{array}$$
It is known from numerical experiments (see [11,12]) that if $c_{0}$ is decreased, the stationary solution of (14) is stable, and if it is very close to −1, the propagation of relaxation oscillations occurs. The value of $c_{0}$ considered here is close to a bifurcation point, which leads to more complex phenomena. Specifically, in this case, we observe the so-called mixed-mode oscillations for a center cell. Relaxation oscillations will propagate at a frequency smaller than the natural frequency of FHN in the oscillatory regime. This is illustrated in Figure 8. In this figure, panels (a), (b), (c) correspond to a fixed value of *x* near the center. At this space location, c(x)=0. Panels (**d**), (**e**), and (**f**) correspond to a value of *x* near the right border. For this latter value of *x*, we have $c\left(x\right)=c_{0}=-1.21$. We observe propagation of oscillations from the center toward the border. Note, however, that only large oscillations propagate. Small oscillations occurring in the center cells are filtered. Panels (**d**), (**e**), and (**f**) suggest the following interpretation of the local dynamics related to wave propagation. In panel (**d**), we represent *u* for fixed *x* as a function of time. We observe at this space location relaxation oscillations at a lower frequency than the oscillatory FHN (i.e., the diffusion-less system for c=0). In panel (**c**), we represent u,v and Δu as functions of time, respectively, with red, dashed purple and blue colors. Through this panel, one can see how the wave propagation is seen locally. It corresponds to a wave of depolarization. Note that the diffusion remains close to zero for almost all times, which corresponds to a solution locally constant in space. When the neighbor cell jumps, the term Δu becomes positive, because at this time we are at a minimum in space. This induces a jump for the considered cell toward the right part of the stable manifold (see panel (**e**)). At this point, the diffusion becomes negative because we are at a local maximum. Then, the diffusion comes back to almost zero again as the considered cell and its neighbors have approximately the same value. The same dynamic occurs symmetrically (the diffusion in this case is first negative and then positive), when the cell goes from the right part of the stable manifold toward the left part of the stable manifold. The dynamics of *v* follows from the fact that $v_{t}=u-c+d_{v}\Delta v$. Panel (**f**) illustrates the dynamics of u,v and Δu in the three-dimensional phase space. The “critical” manifold $v=f\left(u\right)+d_{u}\Delta u$ is also represented where Δu is considered as a variable. It illustrates the oscillations of relaxation for the cells near the border. Illustrations in panels (**a**), (**b**), (**c**) are analog. For this cell the dynamics resemble MMOs often pictured in three dimensional systems.

#### 4.2.2. Fade of Wave Propagation (Death Spot)

In this paragraph, we discuss another phenomenon arising as we vary a parameter (denoted by *p* in this paragraph). We present here the numerical simulation of system (14) with the following set of parameters
(18)$$\begin{array}{c} n=1,\Omega =\left(\alpha ,\beta \right),d_{u}=d=1,d_{v}=0,b=0,f\left(u\right)=-u^{3}+3u\\ I(x)=0\\ c\left(x\right)=p{\left(x/\beta \right)}^{4}-2p{\left(x/\beta \right)}^{2}\\ \beta >0,\;\alpha =-\beta \end{array}$$

In this case, again, as in the previous paragraph, if *p* is close enough to −1, waves propagate from the center at the frequency of the diffusion-less system (with c=0). On the other hand, if *p* is decreased enough, the system generally converges to a stationary solution. In between, for some range of *p*, regular waves propagate from center but fail to propagate at some point in space. The existence of Hopf Bifurcation has been proved for this specific case, see [12]. This type of failure of propagation, also referred as death-spot has been described previously, see for example [52]. For relevance in biological context, we refer to [10]. Related phenomena have also recently been considered in chains of kicked FHN neurons, see [42]. The dynamics under consideration are illustrated here in Figure 9. At the center, oscillations are as in the ODE diffusion-less system. But at some space location, the oscillation fails to propagate. For some intermediate cells, we observe alternations between medium and larger oscillations. In this case, every other oscillation, the amplitude will be shortened— note the difference in the $u_{xx}$ time series. In this figure, panels (**a**), (**b**), and (**c**) correspond to a fixed value of *x* near the center. At this space location c(x)=0. Panels (**d**), (**e**), and (**f**) correspond to an intermediate value of *x* between the center and the left border. In panels (**a**) and (**d**), we represent *u* for fixed *x* as a function of time. In panels (**b**) and (**e**), we represent u,v and $u_{xx}$ as functions of time, respectively, with the colors red, dashed purple and blue. In panels (**c**) and (**f**), we illustrate the dynamics of u,v and $u_{xx}$ in the three-dimensional phase space. The “critical” manifold $v=f\left(u\right)+du_{xx}$ is also represented where $u_{xx}$ is considered as a variable. Figure 10 illustrates *u* as a function of time. From left to right, we represent four different space locations with the same range of amplitude: left panel corresponds to a center cell, the far right corresponds to a cell close to the left border, the two other panels correspond to intermediate cells. This illustrates the fade of the wave propagation. Note the difference of these complex oscillations as compared to the MMOs described in the previous paragraph.

## 5. Discussion

Dynamical systems have proven to be a crucial tool in the description of phenomena observed in neuroscience. This includes complex oscillations, rhythms, wave propagation and pattern formations. As discussed in the introduction, models such as the FHN equations have been used successfully to visualize the emergence of those phenomena and highlighted new ways of biological interpretation. For example, cardiac tachycardia has been associated with spiral waves, which are a classical phenomena studied in FHN systems. Yet, the mechanisms inducing the various dynamics are far from being well-understood. It is necessary to gain a better understanding of the mathematical principles that govern those dynamics to be able to reproduce the biological observed phenomena. In the present article, we focused on the analysis of the FHR and Nh-FHN models. We have provided original theoretical and numerical tools to understand the emergence of complex oscillations and their propagation across excitable media. This included an original theoretical and numerical description of small oscillations and canards leading to different kind of MMOs for the FHR system and an original description of local MMOS, filtering frequencies and fade of wave propagation for Nh-FHN. A better qualitative analysis of the various complex oscillations and their propagation arising in Nh-FHN is an entire topic of research. One possible fruitful way to investigate is to work on finite-dimensional subspaces in which solutions effectively lie. Such insights can be applied to better simulate wave propagation, patterns and better understand their failure and their complexity in the context of neuroscience and biology.

## 6. Conclusions

In this article we have considered the FHR and Nh-FHN models, discussed their relevance in neuroscience context and provided a theoretical and numerical analysis of complex oscillations and their propagation. Future work perspectives include a more detailed investigation of reduction of the Nh-FHN dynamics to ODE models and their various applications to neuroscience and biology. PDEs are infinite dimensional systems; however, asymptotically, the solutions may lie in an attractor, which can be of a much smaller size. Some preliminary unpublished results, see also [13], indicate that the complex oscillations observed in Nh-FHN might be obtained with finite-size ODE models. This is an interesting perspective that would allow a better understanding of the dynamics observed and their biological interpretation.

## Figures and Tables

**Figure 1 biology-12-00918-f001:**
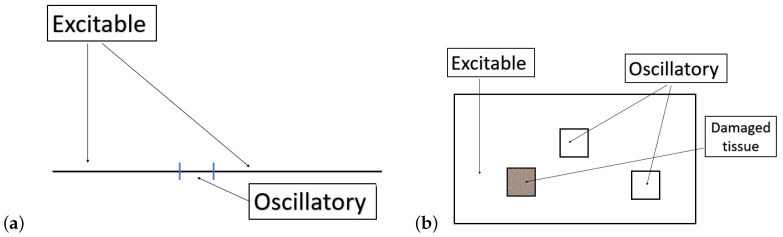
Diagram illustrating the principle of Nh-FHN. FHN ODE models can be excitable or oscillatory depending on a parameter value. Adding a space variable and a diffusive term, one can consider domains where the parameter is space-dependent, opening the possibilities for very rich dynamics. In the left panel (**a**), we illustrate a 1D domain with an oscillatory part at the center and excitable tissue elsewhere. In the right panel (**b**), we illustrate a 2D domain with oscillatory parts, a damaged part and excitable tissue elsewhere. The damaged part can be drawn randomly with a parameter value ranging from values corresponding to very low excitability to high excitability or oscillatory dynamics.

**Figure 2 biology-12-00918-f002:**
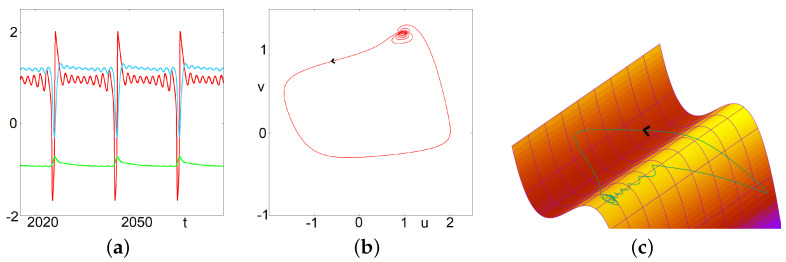
Mixed Mode Oscillations. This figure illustrates the asymptotic behavior of solutions of Equation (2) for I=1.45. (**a**) In the left panel, time series of u,v and *w* are presented. (**b**) In the center, we illustrate the projection of the same trajectory on the plane (u,v). (**c**) In the right, we illustrate the trajectory in the three dimensional phase space along with the critical manifold v=f(u)+w+I. This figure illustrates classical MMOs related to the focus nature of the fixed point of the first two equations (considering *w* as a fixed parameter). In this case, the observed dynamics suggest the following successive states: attractive focus, repulsive focus, relaxation oscillations and return mechanism.

**Figure 3 biology-12-00918-f003:**
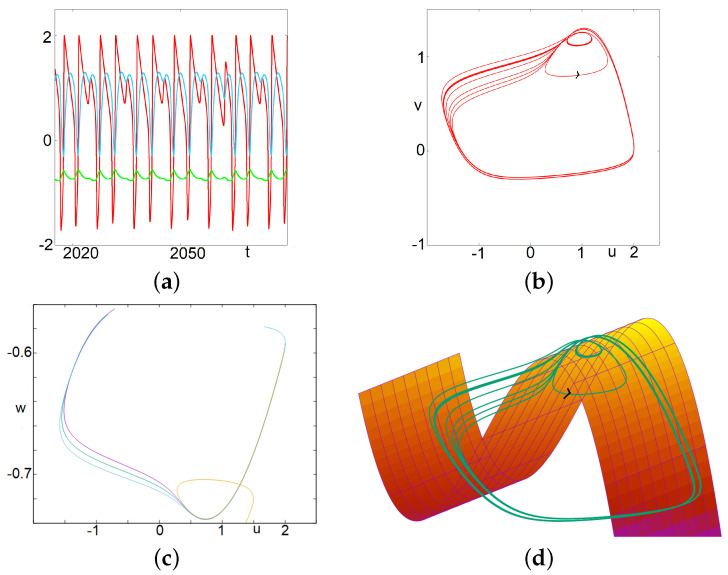
(Complex) Mixed Mode Oscillations. In this case, there are no multiple small focus-like oscillations. Instead, after a relaxation oscillation, the trajectory may enter a canard-type trajectory after crossing the apex of the right part of the critical manifold; it follows the unstable manifold and leaves it either to the left side—where it reaches the left part of the attractive critical manifold—or to the right side—and reaches the right part of the attractive manifold. The latter case corresponds to a small (in fact a middle) oscillation, while the first case gives a large relaxation oscillation. It is worth describing in more detail the trajectory represented in this figure. Two large oscillations are followed by a single small oscillation where the trajectory follows the repulsive manifold before being attracted by the right side of the attractive manifold. This is repeated four times, and the fifth time, the second large oscillation is replaced by a middle-size oscillation. This corresponds to a change in the trajectory occurring in a tiny value range (see bottom left). After that, the cycle is repeated. Introducing the letter M for the medium oscillation, L for large and S for small, we could denote this occurrence: LLSLLSLLSLLSLM. (**a**) time series of *u*, *v* and *w*. (**b**) projection on the u−v plane. (**c**) projection on the u−w plane of four successive segments of the trajectory along the repulsive branch and the subsequent fast trajectory. The first three segments exit towards the left part of the attractive manifold. The fourth one exits toward the other side. (**d**) trajectory in the three dimensional phase space along with the critical manifold v=f(u)+w+I.

**Figure 4 biology-12-00918-f004:**
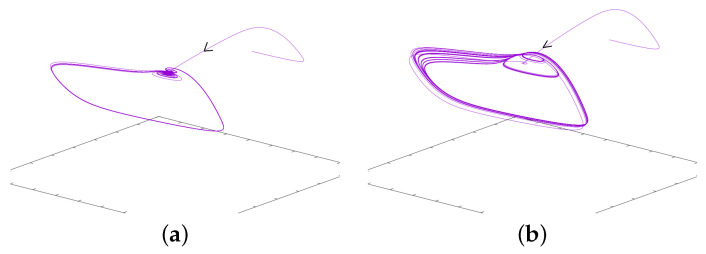
This figure illustrates a remarkable transient behavior. For the parameter values considered here, the stationary point has two complex eigenvalue with positive real parts and one negative real eigenvalue. Accordingly, for the initial conditions chosen here, the trajectory follows the stable manifold associated with negative eigenvalue. After that, the trajectories are repelled from the fixed point. The asymptotic behavior, however, does not relate so much with those eigenvalues. As such, the Shilnikov chaos is not relevant for these parameter values. (**a**) I = 1.45. (**b**) I = 1.3.

**Figure 5 biology-12-00918-f005:**
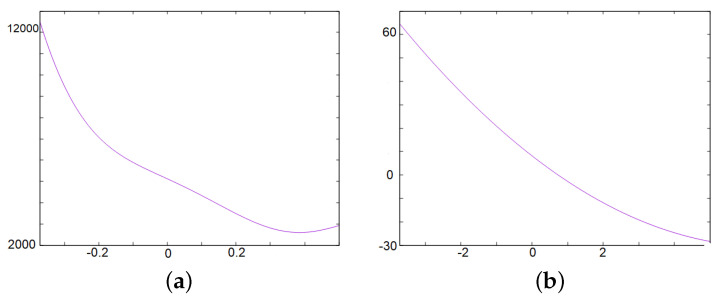
This figure gives an illustration of Proposition 3. In both panels the x–axis represents $f^{\prime }\left(u^{*}\right)$ for I∈[1,2]. (**a**) illustration of −Δ as defined in the Cardan formula. Since Δ is negative, the determinant admits two complex eigenvalues. (**b**) illustration of $a_{2}a_{1}-a_{0}$ as defined in the Routh–Hurwitz criterion. It is positive for $I>I^{*}$. We also have $a_{0}>0$ and $a_{2}=\epsilon +b-\frac{f^{\prime }\left(u\right)}{\epsilon }$, which is positive for $I>I^{*}$, ($I^{*}$ is the first value for which $a_{2}a_{1}-a_{0}$ or $a_{2}$ equals zero).

**Figure 6 biology-12-00918-f006:**
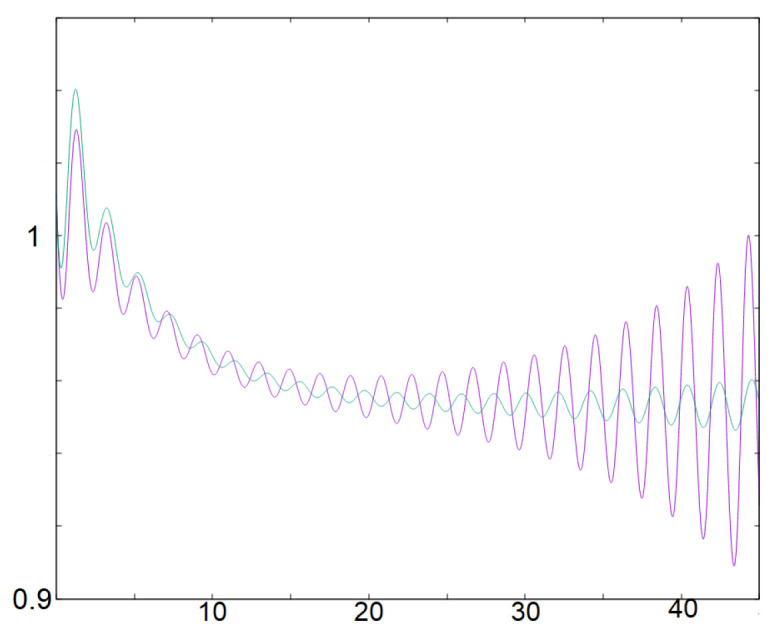
This figure illustrates small oscillations for systems (9), in green, and (2) in purple. At the beginning the solutions are very close, but since we dropped the nonlinear terms the solutions are distinct after some time. Note, however, that the solutions of (9) capture two aspects of the small oscillations occurring for (2). First, it allows to generate and control in a simple way small oscillations by mimicking the behavior near a focus. Next, it generates oscillations that are first decreasing and then increasing, which corresponds to a dynamical Hopf-bifurcation behavior (TrA changes its sign and ${\left(TrA\right)}^{2}-4\det A<0$ ).

**Figure 7 biology-12-00918-f007:**
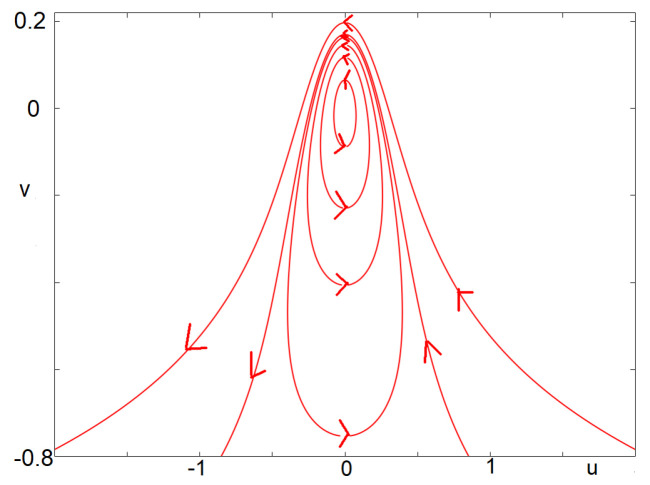
Solutions of the integrable Equation (13). The fixed point lies in the middle of limit cycles (nonlinear center). Above some threshold, solutions are no longer periodic but rather escape the vicinity of the stationary point. This illustrates the behavior of Figure 3: some canards exit to the left, while others exit to the right.

**Figure 8 biology-12-00918-f008:**
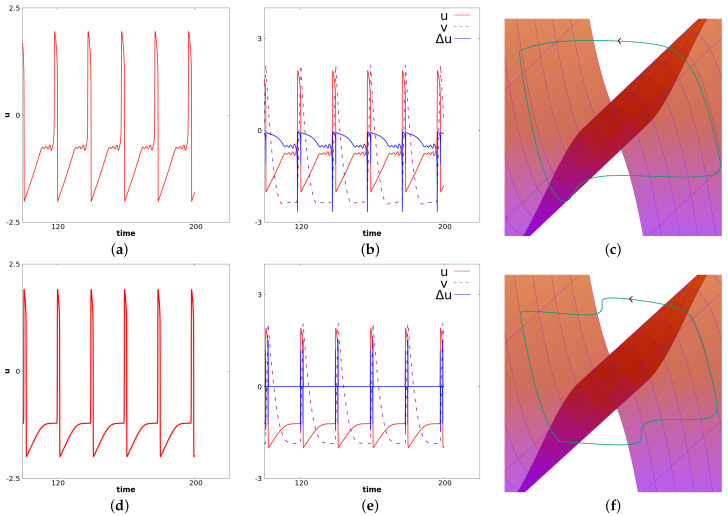
Simulation of Equation (14) with the set of parameters given in (17). The first row corresponds to a fixed value of *x* near the center for which c(x)=0. The second row corresponds to a value of *x* near the left border for which $c\left(x\right)=c_{0}=-1.21$. In the first column, we represent *u* as a function of time. In the **second column**, we represent u,v and Δu as functions of time, respectively, in green, blue and violet. In the **third column**, we represent the trajectory u,v,Δu along with the manifold $v=f\left(u\right)+d_{u}\Delta u$ (where Δu is seen as the third variable). These figures illustrate the apparition of MMOs for center cells and the propagation of relaxation oscillations from the center cells toward the border with a filtration of small oscillations.

**Figure 9 biology-12-00918-f009:**
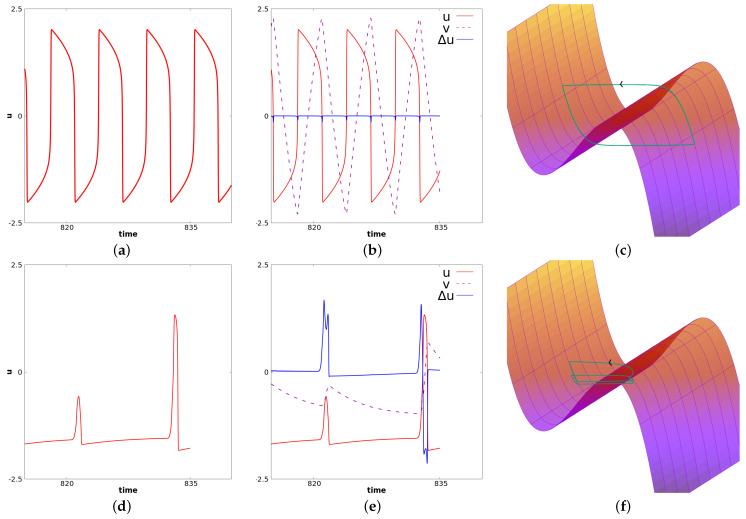
This figure illustrates the simulation of Equation (14) with the set of parameters given in (18). Panels (**a**–**c**) correspond to a fixed value of *x* near the center for which c(x)=0. Panels (**d**–**f**) correspond to a value of *x* at an intermediate location between the left border and the center. In (**a**) and (**d**), we represent *u* as a function of time. In (**b**) and (**e**), we represent u,v and $u_{xx}$ as functions of time, respectively, in green, blue and violet. In (**c**) and (**f**), we represent the trajectory $u,v,u_{xx}$ along with the manifold $v=f\left(u\right)+du_{xx}$ (where $u_{xx}$ is seen as the third variable).

**Figure 10 biology-12-00918-f010:**
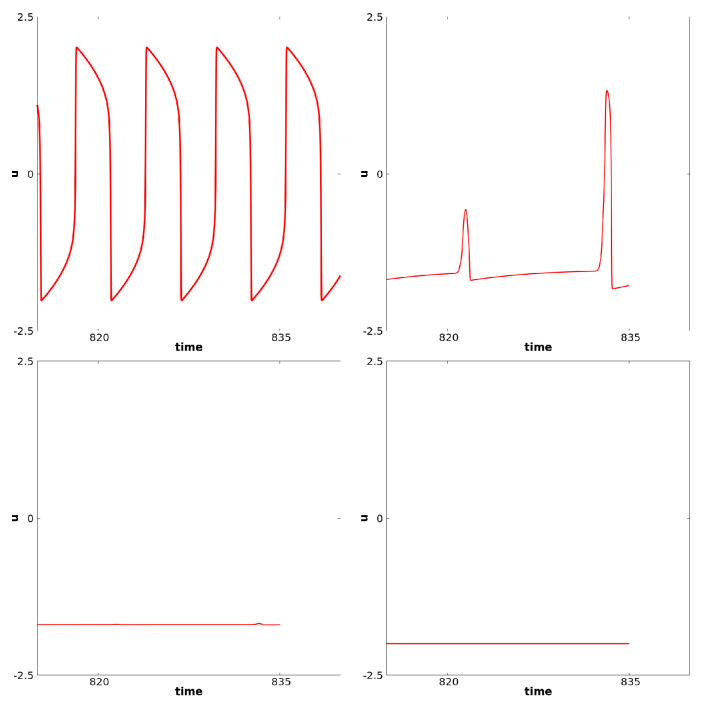
This figure illustrates the fade of the wave propagation. We represent here the value of *u* as a function of time. From left to right, we consider four different space locations with the same range of amplitude: left panel corresponds to a center cell, the far right corresponds to a cell close to the left border, the two other panels correspond to intermediate cells.

## Data Availability

Not applicable.

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
