# Peer review of "Non-Trivial Dynamics in the FizHugh–Rinzel Model and Non-Homogeneous Oscillatory-Excitable Reaction-Diffusions Systems"

_biology, 2023, doi:10.3390/biology12070918_

Round 1

Reviewer 1 Report

This work provides another example of mixed-mode-oscillations in three-time-scale systems, which was examined initially by Krupa et al in Ref 37. Technical part is clearly described but I suggest a better organization of the abstract and the introduction. There are lots of repetitions and it is hard to follow.

This very theoretical work is without a clear motivation of the biological context and application. It is not very clear to me why the authors had chosen Biology, rather than SIADS, Chaos or the Journal of Mathematical Biology, etc., where the theoretical work would be more appreciated. If the authors insist on their choice, they should be a lot more precise for putting their work in a biological context rather than leaving to work to the reader, as I exemplify below:

- pg 1 lines 15-16: “the emergence of complex oscillations arising in a few dynamical systems typical in neuroscience.” (Please exemplify clearly.)

- pg 16 lines 375-376: “For relevance in biological context, we refer to [12].” (it is not the reader who should look for context but the author.)

Mixed-mode-oscillations have already been reported in different FHN systems. For instance, the authors may want to compare their work with “Jone Uria Albizuri, Mathieu Desroches, Martin Krupa, Serafim Rodrigues. Inflection, Canards and Folded Singularities in Excitable Systems: Application to a 3D FitzHugh–Nagumo Model. Journal of Nonlinear Science, 2020, 30 (6), pp.3265-3291.” Also, the recent works on three-time-scale systems should be cited appropriately.

Authors may want to add a conclusion section, even very short.

Pg 1 line 27: in general, it is not neuron’s axon that is excitable. The role of the axon is to propoage an action potential generated at the level of the axon initial segment. Thus it is the soma that is excitable.

Can be improved.

Author Response

We want first to thank the reviewer for his review and his valuable comments. We would like to confirm our choice to publish in Biology. Following the first remarks of the reviewer, we have substantially detailed the biological context: in particular, in addition to an extended more detailed general context presented in the introduction, we have also detailed the content of the article of Maia and Kutz which is relevant for our article (in the introduction) , and added several biological references where complrx oscillations such as mixed mode oscillations and bursts are relevant in section 2.1. We have also mentioned the article suggested by the reviewer on inflection sets as well as articles with three time scales. For the last comment of the reviewer, we agree. We want however to clarify our choice of vocabulary as follows. When an action potential is propagated along the axon, this occurs thanks to a combination of diffusion and ions fluxes. That was a major success of HH equations and subsequent excitable diffusive systems such as FHN. And that is exactly the point of excitability in FHN. The system is stable, but it can go through a relaxation oscillation if it crosses a threshold; in this case the gates are hidden in the recovery variable v of FHN, but it is what is commonly called an excitable system. In our Nh-FHN model, the soma could be for example represented by an oscillatory segment and the stationary (excitable) segments would represent the axon as its ability to propagate the original oscillations thanks to diffusion and ion fluxes. Other more complex interpretations are also possible such as damaged zones in excitable tissues.

Reviewer 2 Report

This paper addresses the slow-fast dynamics of a FitzHugh type 3D model, considered as a toy model of the more complex Hdgkin-Huxley model. They reviewed the slow-fast analysis of the 3D system, they performed the numerical analysis and found mixed-mode oscillations and canard solutions. Then they numerically analyse a reaction-diffusion FitzHugh model near the mixed mode oscillation regimes. The paper presents new results for the considered dynamical systems and is well written. 

Author Response

We thank the reviewer for his comments.

Reviewer 3 Report

The article focuses on the qualitative analysis of complex dynamics in mathematical models used in neuroscience applications. Specifically, it discusses two types of models: the 3-dimensional FitzHugh-Rinzel (FHR) model and a class of non-homogeneous FitzHugh-Nagumo (Nh-FHN) Reaction-Diffusion systems.

The FHR model and Nh-FHN models are relevant in the context of neuroscience because they can generate complex dynamics and wave-propagation phenomena. These dynamics include canards, Mixed-Mode Oscillations (MMOs), Hopf-Bifurcations, and their spatially extended counterparts.

The article presents original methods to characterize these complex dynamics and explains how they emerge in both ordinary differential equation (ODE) models and spatially extended models. It likely discusses techniques such as bifurcation analysis, numerical simulations, and mathematical tools to study the behavior of these models.

The research topic is relevant, but the article needs to be revised:

1. The observed regimes may be of interest from a mathematical point of view, where the boundaries of the model are not defined from a physical point of view. What does your research contribute to understanding the biological processes of such dynamics? References to biological experimental data are required.

2. A graph with a model diagram is required.

3. The description of the model must be moved to the methods section. Also, in the methods section, it is necessary to add the values of the parameters of the systems under consideration, as well as the methods of numerical and bifurcation analysis.

4. All values of the axes in figures and captions must be legible. It is necessary to increase the font size in all figures.

5. Check what you have on lines 191, 201, 266, 314 (squares are displayed).

6. It is necessary to add Discussion and Conclusion Sections.

Moderate editing of English language.

Author Response

We thank the reviewer for his valuable comments.

  1. We have considerably developed the introduction to add more biological context. We have in particular detailed the main point of the article of Maia and Kutz which is to distinguish between deformations of axons which do not affect spike propagations and those which induce failure of propagation. Our article presents indeed a detailed analysis of the dynamics which can help to understand and model in more detail the propagation and failure of action potentials. More complex interpretations are also relevant such as damaged tissues.
  2. We added a graph see Figure 1.
  3.  We used our own C++ program for the numerical simulations. We added such a statement in the article. We also propose to add a repository on GitHub with the programs and to put the link on the article. Since the article presents substantial detailed mathematical content, we would prefer to keep the structure and presentation of the models as it is.
  4.  We totally agree with the reviewer's comments on figures. For timeline efficiency we propose to include the new version of figures with readable legends for the final round.
  5. We added conclusion and discussion sections.

Round 2

Reviewer 1 Report

Authors addressed my concerns.

Author Response

Thank you.

Reviewer 3 Report

1. In the presented version of the manuscript, changes from the original version of the manuscript are not highlighted, which makes it difficult to evaluate its revision. It is necessary to highlight all the adjustments that have been made. In the current version, I don't see any changes in line with the previous remarks, except for the addition of a Conclusion and Discussion section, i.e. comments 1 to 5 were simply ignored and not used to refine the article. I expect improvements on all previous comments.

2. Incorrect order of the Conclusion and Discussion sections is specified (the Discussion section comes first, then the Conclusions). In the Conclusions section, it is necessary to expand the possible directions for further work. The Discussion section should be greatly expanded. On line 455, the sentence appears to be unfinished.

Perhaps it makes sense for the authors to consider another journal more suitable for theoretical work, such as Mathematics.

In this form, the article cannot be accepted.

Moderate editing of English language required

Author Response

  1. We thank the reviewer for his patience and his remarks. It is correct that the corrections were not highlighted in the previous revised version. They are now, and we will also comment some of them hereafter. In comparison with the original version, we have substantially developed the introduction to include more biological references and details. We explain how network models for the fruit fly and the visual cortex have been successfully used to provide important contributions in Neuroscience. We also mention how Non Autonomous Poisson equations have been used to provide insights about interpretation of EEG recordings in the context aphasia. In those papers, rhythms, oscillations, and waves play a crucial role. We also added references of the use of the FHN model in the context of tachycardia and gave more detail about a 1d FHN model discussing action potential propagation failure.  We believe that the new version of the article addresses the previous comment 1 of the reviewer. We have added a diagram (Fig. 1) to clarify one of the key ideas of the manuscript. It is our understanding that this addresses the previous comment 2.  The description of the models appears now in the methods section. We also added the values there. We wrote our own programs in C++ and used a RK4 method in time with finite difference in space. This is mentioned in the method section.  We changed the font size in the figures. The squares indicate the end of mathematical proofs.
  2.  We expanded the discussion and the conclusion and reordered it.

Round 3

Reviewer 3 Report

Thanks to the authors for their responses to comments.